# LipidCreator workbench to probe the lipidomic landscape

Bing Peng[1,2,18], Dominik Kopczynski[1,18], Brian S. Pratt[3], Christer S. Ejsing[4,5], Bo Burla[6], Martin Hermansson[4,7], Peter Imre Benke[8], Sock Hwee Tan[9,10], Mark Y. Chan[9,10,11], Federico Torta[8], Dominik Schwudke[12,13,14], Sven W. Meckelmann[15], Cristina Coman[1,16], Oliver J. Schmitz[15], Brendan MacLean[3], Mailin-Christin Manke[17], Oliver Borst[17], Markus R. Wenk[6,8], Nils Hoffmann[1] & Robert Ahrends[1,16 ✉]

Mass spectrometry (MS)-based targeted lipidomics enables the robust quantification of selected lipids under various biological conditions but comprehensive software tools to support such analyses are lacking. Here we present LipidCreator, a software that fully supports targeted lipidomics assay development. LipidCreator offers a comprehensive framework to compute MS/MS fragment masses for over 60 lipid classes. LipidCreator provides all functionalities needed to define fragments, manage stable isotope labeling, optimize collision energy and generate in silico spectral libraries. We validate LipidCreator assays computationally and analytically and prove that it is capable to generate large targeted experiments to analyze blood and to dissect lipid-signaling pathways such as in human platelets.

[1] Leibniz-Institut für Analytische Wissenschaften – ISAS - e.V., 44139 Dortmund, Germany. [2] Division of Rheumatology, Department of Medicine Solna, Karolinska Institutet, Karolinska University Hospital, SE-171 76 Stockholm, Sweden. [3] University of Washington, Department of Genome Sciences, WA 98195 Seattle, USA. [4] Department of Biochemistry and Molecular Biology, University of Southern Denmark, DK-, 5230 Odense, Denmark. [5] Cell Biology and Biophysics Unit, European Molecular Biology Laboratory, 69117 Heidelberg, Germany. [6] Singapore Lipidomics Incubator (SLING), Life Sciences Institute, National University of Singapore, 117456 Singapore, Singapore. [7] Wihuri Research Institute, 00290 Helsinki, Finland. [8] Singapore Lipidomics Incubator (SLING), Department of Biochemistry, Yong Loo Lin School of Medicine, National University of Singapore, 117596 Singapore, Singapore. [9] Department of Medicine, Yong Loo Lin School of Medicine, National University Hospital, 119228 Singapore, Singapore. [10] Cardiovascular Research Institute, National University of Singapore, 117599 Singapore, Singapore. [11] National University Heart Centre, National University Health System, 117599 Singapore, Singapore. [12] Research Center Borstel, Leibniz Lung Center, Borstel, Germany. [13] German Center for Infection Research (DZIF), 38124 Braunschweig, Germany. [14] Airway Research Center North Member of the German Center for Lung Research (DZL), 22927 Großhansdorf, Germany. [15] Applied Analytical Chemistry, University of Duisburg-Essen, 45141 Essen, Germany. [16] Department of Analytical Chemistry, University of Vienna, Währinger Strasse 38, 1090 Vienna, Austria. [17] Department of Cardiology and Cardiovascular Medicine, University of Tübingen, 72076 Tübingen, Germany. [18] These authors contributed equally: Bing Peng, Dominik Kopczynski. ✉email: robert.ahrends@univie.ac.at

Life is swaddled in lipids; they form cells and organelles, mediate information flow, protect cells and tissues from a hostile environment and serve as energy building blocks. Imbalanced lipid homeostasis leads to numerous diseases[1–7]. Consequently, lipids are of great clinical interest and have high potential as biomarkers[8,9]. However, their chemical structure is not linearly constructed, as it is for DNA, RNA and proteins, which form a sequence of nucleotides and amino acids. Lipids are chemically highly diverse and are assembled from combinations of distinct building blocks comprising backbones, head groups, sugars and fatty acyls with different lengths, numbers of double bonds and bond types. As mass spectrometry (MS) has gained speed and sensitivity, it has facilitated lipid identification by deciphering their chemical components and structure[10–12], thereby helping to establish the emerging field of lipidomics.

Despite the rapid growth of the field, there is a lack of a comprehensive software toolbox for lipidomics, which focuses on the reproducible, quantitative analysis of a subset of lipids of interest. MS techniques such as selected reaction monitoring (SRM) or parallel reaction monitoring (PRM) are employed for lipid quantification[13,14]. However, such targeted analyses widely applied in proteomics have led to software developments such as Skyline[15,16], which was further advanced by tools such as Panorama, and MSStats[17,18]. Currently, no open source tool offers a user-friendly interface for the development of targeted lipidomics assays that addresses the structural complexity of eukaryote lipidomes. However, there is a strong need in the field of metabolic research and lipid signaling to target distinct sets of molecules in line with their associated biological function, in contrast to screening approaches. In a previous study[19], we demonstrated the usability of Skyline for lipidomics by encoding lipids as pseudo-peptides and their fragments as pseudo-amino acid residues. However, this approach was not broadly applicable; it worked only for a small subset of lipids and could in retrospect only be an interim solution.

To quantify diverse lipid classes and lipid-signaling molecules in large cohorts, targeted workflows should be quick to establish, and the obtained results should be easy to inspect and validate. In reality, this process is laborious, time consuming and most often not accessible to non-experts. To foster application of mass spectrometry based lipidomics on the individual level of experience, we here present a targeted lipidomics workbench and lipid knowledge-base that is fully integrated with Skyline.

## Results

**Features and integration**. To address the challenges mentioned above, we introduce LipidCreator for the automated generation of targeted lipidomics MS assays (Fig. 1a,b). Assay generation can be conducted with a graphical user interface (GUI) or by using command line functionality, covering lipids of the following categories: sphingolipids (SP), glycerolipids (GL), glycerophospholipids (PL), sterol lipids (ST), and fatty acyls including mediators (LM) (Supplementary Data 1). LipidCreator can calculate mass to charge ratios ($m/z$) for lipid species and their derived fragment ions, covering over 61 lipid classes and a lipid array of $10^{12}$ lipid molecules (Supplementary Table 1, Supplementary Data 2, Fig. 1a(1)-(3)). The fragmentation information is obtained from literature and own fragmentation experiments[13,20–30] (Supplementary Table 2). On this basis, the computational permutation of precursors and fragments considering double bonds and chain length is carried out to calculate the present lipid array (Supplementary Table 3). From this array, the lipids of interest are selected with the consensus nomenclature recommended by the Lipidomics Standards Initiative (https://lipidomics-standards-initiative.org)][1,31] (Supplementary Table 4,

Supplementary Datas 3 and 4, Fig. 1a(4)&(6)). The stable isotope feature of LipidCreator enables the custom creation of labeled lipids and their transitions (Fig. 1b(3)). This simplifies their inclusion as internal standards for the validation and quantification of lipid species in assays by stable isotope dilution MS or for the tracing of lipid building blocks by FLUX analysis.

In addition to the computation of targeted assays, an in silico spectral library can be generated (Fig. 1a(7), b(5)). Besides the generation of fragment ions, a critical feature is the ability to determine the relative intensities of fragment ions at different defined collision energies (CE). We therefore trained nonlinear regression models on empirical data from measurements of lipid mediator standards on two different MS instrument types (Supplementary Note 3). These fragment ion-specific models provide the link between CE and predicted relative intensity (Supplementary Figure 52) that enables us to generate a spectral library and an optimized CE-based transition list (Fig. 1b(5)& (6)). The spectral library can be used to support the development of targeted assays by simplifying the validation of all obtained fragment ion traces from SRM, PRM or DIA by matching their mass-spectral fingerprint against the library's spectra using the dot product in Skyline (Fig. 1a(9), b(7)). This facilitates the identification of individual lipid species with high confidence if the relative pairwise ion ratios are sufficient for distinction. The CE optimization is also very useful if individual precursor fragment transitions are being established such as for SRM assays using triple quadrupole MS.

Supplementing the fully accessible workbench, we also prepared predefined transition lists for several model system lipidomes, such as yeast, mouse (platelets, heart, brain hippocampus), drosophila and human (plasma, platelets)[32–36], making it straightforward to start a targeted lipidomics experiment.

Using its stand-alone batch-processing mode to create transition lists from lipid names, it computes precursor-product ion pairs at a rate of 60,000 pairs per second on a standard notebook (i5-4310M CPU @ 2.70 GHz with 8 GB of main memory). LipidCreator can be integrated into KNIME workflows as an external tool node via its command line interface on Linux and Windows. It is also a native Skyline plugin (Skyline 64 bit, version 19.1 and above), which greatly facilitates vendor-independent assay development and analysis, data visualization and quality control of MS and MS/MS data (Fig. 1c). LipidCreator further supports visual inspection of fragments (Supplementary Note 2) and a lipid name translator supporting LIPID MAPS nomenclature[37].

**LipidCreator design and architecture**. LipidCreator is written in the C# language and can be run on both MS Windows (stand-alone or as a Skyline plugin) and on Linux (using the Mono framework). It contains multiple levels of structured data to create lipid transition lists (Fig. 2). The first level, which is the precursor level, contains information about all lipid classes. This information refers to the precursor m/z computation. Information such as lipid category, lipid class name, length of fatty acyl chain, type of fatty acyl chain and adduct attributes are included. In the user interface, the lipid backbone indicates how the according lipid is assembled (for instance a cardiolipin contains four fatty acyls etc.). The user has to define the various global parameters of the lipid (head groups, adducts, fatty acyls or long chain bases), where each lipid class contains individual computational rules to assemble head groups, fatty acyls and enabled adducts to create a customized precursor group. To avoid redundancy (like for PC 12:0–14:0 and PC 14:0–12:0), the fatty acyls are sorted numerically according to carbon length, number of double bonds and number of hydroxyl groups. Each precursor

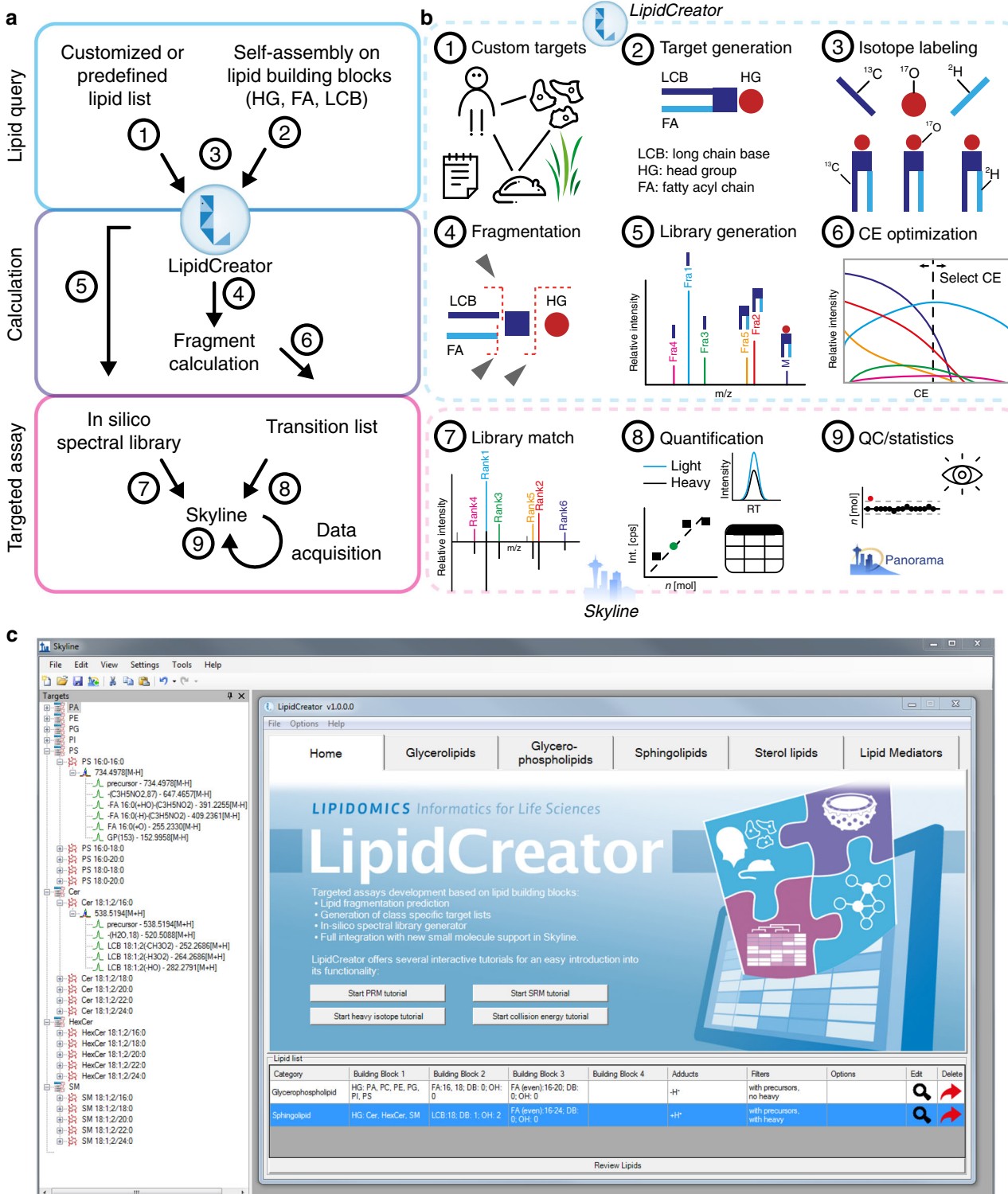

**Fig. 1 The LipidCreator workbench and its integration into Skyline. a** Main steps include the decision of which lipids to query (Lipid Query), the calculation of the lipid precursor and corresponding fragment masses (Calculation) and the assay design (Targeted Assay). **b** These main steps consist of the selection of species or lipidomes to target (1–2), the inclusion of isotope-coded internal standards for validation (3), the calculation of precursor and fragment masses for the assay (4), library generation (5) and the in silico optimization of CE for individual fragments (6). The steps (1–6) are performed in LipidCreator. After the submission of the transition list to Skyline and the execution of the targeted lipidomics experiment, lipids can be validated by spectral library matching (7) and/or their coeluting internal standards, which are further used for quantification (8). Due to the integration features of Skyline, different downstream quality control systems, such as Panorama, are available (9). **c** Graphical user interface of LipidCreator integrated into Skyline.

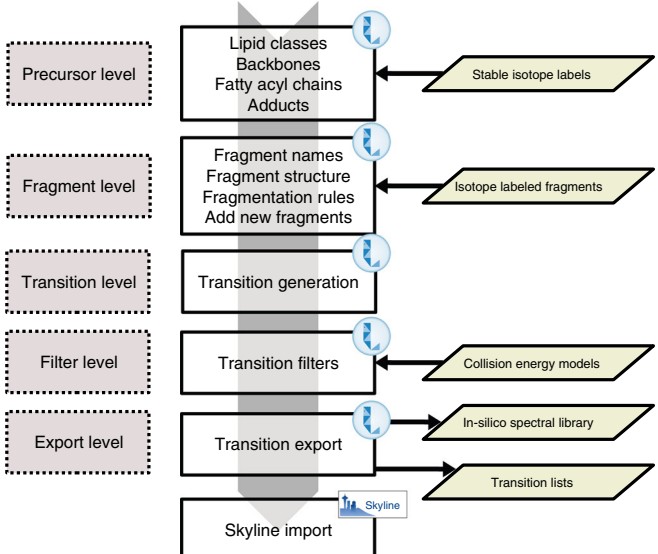

**Fig. 2 LipidCreator system levels and flow of information.** LipidCreator uses an internal knowledge base that stores information about lipids and their fragments. This information is assembled based on the user's selection of lipid classes and additional parameters from the two main levels, the precursor and fragment level. Then, LipidCreator combines the user's selection with information about lipid classes, precursors, backbones, fatty acyl chains, lipid fragments and information on isotope labels and forwards it to the transition layer for targeted assay generation. The downstream filter layer then applies collision-energy models to the generated transitions, if either automatic CE optimization or manual CE mode are enabled. The final export layer generates the final transition lists and spectral libraries and either stores them locally or transmits them directly to Skyline.

group contains all the information of its lipid building blocks to simplify computation of fragment-specific information.

The fragment level is the second level and stores information such as the fragment names, the corresponding structure images and calculation rules which are later utilized to generate the masses of MS fragments of interest. Users can manually define additional fragments that are not provided by LipidCreator using this level. We support both positive and negative mode for each lipid class. Finally, the exported transition list includes all mandatory information for both precursor and fragments, such as name, mass, adduct, and charge state.

Both levels use additional definitions for heavy isotope-coded precursor and fragment ions following the aforementioned rules. Finally, if enabled, we apply the collision-energy models to calculate the relative fragment intensities based on the automatically calculated optimal collision energies or based on the manually provided ones to generate the in silico spectral library for use by Skyline or other tools that support the Bibliospec blib format.

**Optimization of collision energy and library generation.** LipidCreator offers a module, enabling the optimization of CEs for a sophisticated lipid analysis of molecular species. Here, the user can choose between two complementary modes, finding the optimal CE of one specific quantifier fragment ion or, defining the CE with the highest information content in MS/MS. Furthermore, all CE choices can be performed manually for the highest level of customization of a dedicated lipidomics assay.

Due to the high diversity of lipid fragment properties, we applied statistical models for the automated CE calculation mode that are based on experimentally acquired data. Figure 3 shows the CE optimization workflow for LipidCreator. To determine

suitable collision energies for lipid classes or species defined in LipidCreator, these lipids need to be selected and added to the Lipid list (Supplementary Note 2) to define a targeted assay (Fig. 3, (1)). Each lipid definition includes a set of fragments that were obtained from literature or were generated based on the building blocks and the parameters set in the LipidCreator user interface. We then exported these lipid target lists as target transition lists. In step (2), we performed repeated experimental MS/MS measurements of the lipids that we selected in (3) on the respective target MS platform with different CEs. For the QExactive HF platform, we acquired the spectra from normalized collision energy (NCE) 10–60, with a step increase of 1 and an average of 16 repetitions per NCE (minimum 1, median 18, maximum 21) and precursor. For the QTOF platform, we acquired the spectra from CE 10–100, with a step increase of 1 and an average of 3 repetitions per CE (minimum 1, median 3, maximum 7) and precursor.

The vendor MS raw data was converted into a centroided format[38] using Proteowizard's MsConvert[39] but applied peak picking algorithm libraries of the vendor. Next, we combined the target transition lists of step (1) with the converted mzML files of step (2) to select and transform $m/z$ values and (relative) intensities into intermediate feature tables (see Supplementary Note 2) in step (3). In step (4), we processed the extracted feature tables using the open source R-package flipR (Fragment-based Lipid Intensity Prediction, see Source code availability). flipR fits multiple nonlinear regression models for each lipid-specific transition fragment within a user-defined parameter range, given the scan-relative intensities over the sampled range of collision energies. We applied flipR to create QC plots (5) to assess potential technical issues, estimated the model parameters and inserted them into LipidCreator to calculate optimized collision energies (6). The automatic collision-energy selection uses the mode of the product density distribution over all individual fragment distributions. Thereby, the selected CE covers the highest simultaneous product overlap over all fragments. The user can customize the preselection of all fragments in LipidCreator, which updates the mode calculation of the product density distribution accordingly (see Supplementary Figure 1). An overview of the total number of samples per fragment and PPM mass error is available in Supplementary Note 3 for 10-HDoHE. Supplementary Data 5 (QExactive-HF) and Supplementary Data 6 (QTOF) report summary information and diagnostic plots for all mediators.

**Computation of lipidome coverage.** To underline the wide applicability of LipidCreator, and to prove the correctness of the generated transitions, we performed computational, analytical, and biological validation experiments for different use cases ranging from the computation of lipidome coverage to their applicability to investigate lipid-signaling pathways.

LipidCreator covers lipid classes that occur in many biological organisms. Several reported lipidome compositions from the literature were used to determine the coverage provided by LipidCreator (Fig. 4) for the following organisms and tissues: (a) Human plasma[40], (b) Human platelet[36], (c) Mouse heart[35], (d) Mouse platelet[36], (e) Mouse brain[32,41], (f) Yeast[34], (g) Zebra-fish[42], (h) Drosophila[33,43], (i) Arabidopsis[44], and (j) E. coli[45–47]. We can show that the vast majority of reported lipids among the selected model organisms are supported by LipidCreator. For expanding the coverage of LipidCreator by adding lipid classes and categories, we encourage the community to open pull requests or issues within our github repository (https://github.com/lifs-tools/LipidCreator). That is how we can ensure a clean inclusion of the data and a stable functionality of the tool.

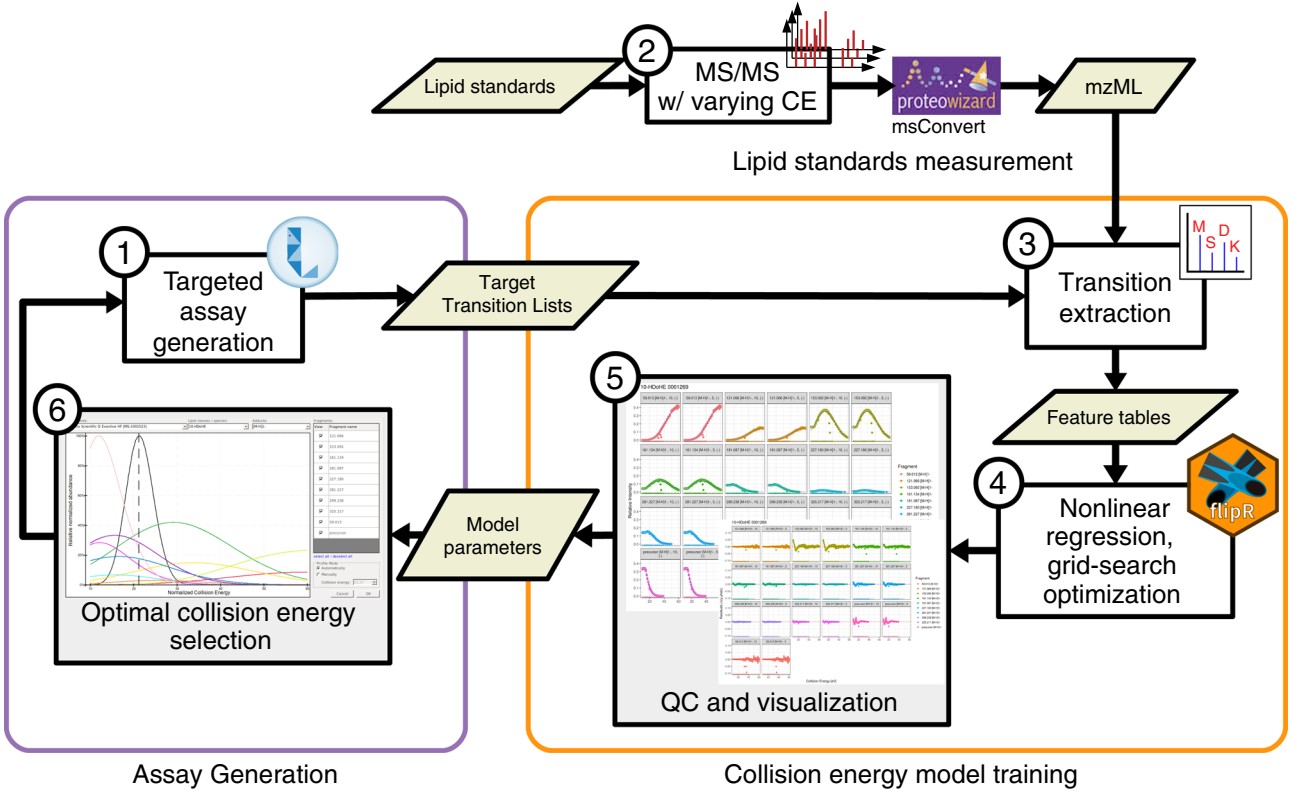

**Fig. 3 Relative fragment intensity prediction for collision-energy optimization.** The workflow is connecting the assay development with LipidCreator (1), lipid standards measurement (2) and transition extraction from the measured data to create feature tables (3) for the training of fragment-specific nonlinear regression models for collision-energy-dependent, relative intensity prediction using nonlinear regression models (4). The subsequent QC and visualization step (5) supports model fit quality inspection and selection of the parameterized model parameters. LipidCreator uses these model parameters to calculate the optimal collision energy for a lipid based on the selected fragments (6) for further assay refinement and integration with the MS acquisition workflow.

**False match and target-decoy calculations.** To validate the accuracy of LipidCreator's transition lists, we conducted two experiments in a well-controlled manner (Fig. 5). The objective of the first, computational experiment (i) was to determine the number of transitions per lipid required for identification within a small and well characterized lipidome. In the second experiment (ii), we validated our theoretical calculations and the transition lists computed by LipidCreator by monitoring true and false targets. Therefore, we depicted the lipidome of the yeast *Saccharomyces cerevisiae* as a matrix. For yeast, it is well known which lipid classes occur, which numbers of fatty acyls and double bond varieties exist, and most importantly, which fatty acyls and lipid classes do not occur[34]. This knowledge is important for our consecutive false identification calculations[48].

We calculated a probability measure for lipid identification. Thereby an important information for lipid identification is the number of fragments per lipid taken into consideration. Obviously, it holds that as more fragments are available the better the identification becomes. Therefore, it is inevitable to have a measure of reliability of the identification with respect to the minimum fragment number necessary for an unambiguous identification at the molecular lipid level[31,49]. However not always all available fragments can be considered due to a limited scanning time. Thus, one has to balance the number of fragments per precursor analyzed and the numbers of precursors monitored. Hence, we performed a theoretical calculation to infer the minimum number of fragments necessary for an unambiguous identification of a molecular lipid species within a certain lipidome. Here, we denote a lipid as unambiguous with respect to a number of n fragments only when (i) no other lipid possesses

a similar precursor mass in this set or (ii) having the same precursor mass, no other lipid contains *n* fragments with similar fragment masses. A fragment's mass is termed similar to a reference mass if and only if their mass difference does not exceed a pre-specified mass tolerance. For our calculation, we used 263 lipid species (367 precursors, 146 negative / 221 positive precursors) from yeast as background matrix[49], 32 lipids (42 precursors, 18 negative / 24 positive precursors) as target lipids, and 38 (52 precursors, 24 negative / 28 positive precursors) as decoy lipids. Note that all lipid species are unambiguous among all three sets. Figure 5a–d illustrate the results. On the MS level, the masses of each precursor were controlled for similarity against all remaining precursor masses. If two precursor masses were similar, the maximum overlap of fragments of a reference lipid with the fragment of its similar precursor was computed. The overlap of two fragment lists was computed by using a version of global sequence alignment adapted to handle numerical values[50].

When identification is solely based on the precursor mass, the probability to unambiguously identify a lipid is 64%. When using one arbitrary fragment, the probability increases up to 94%, while using two arbitrary fragments leads to a value of 97%. Therefore, especially on low resolution devices, we recommend to verify the target lipid with at least two fragments during the initial method development (Fig. 5c,d). By monitoring more than three transitions, the false match probability can be reduced to 2% for all lipid species.

In addition, we chose a target-decoy approach to validate the performance and accuracy of the transition lists we created with LipidCreator. Therefore, we selected 21 target lipids and 21 decoy lipids from the yeast lipidome matrix that we calculated

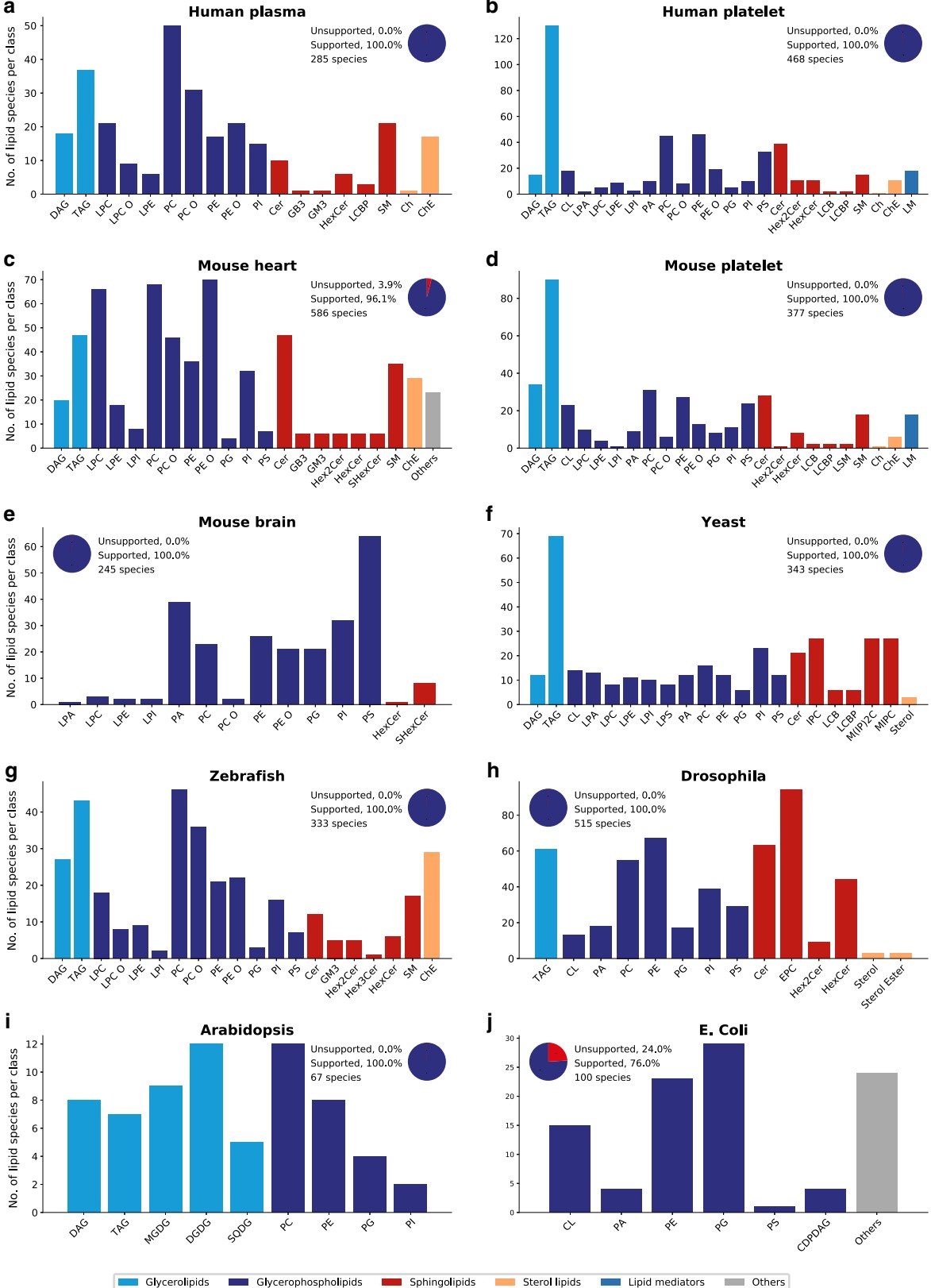

**Fig. 4 Lipid distribution and lipidome coverage of LipidCreator in different model organisms. a–j** Numbers of lipid species per lipid class within different organisms. A list with all unsupported lipid classes within this experiment is available in Supplementary Note 1 and in Supplementary Data 1. The latter table further contains detailed descriptions of all lipid name abbreviations and the lipid classes supported by LipidCreator. Source data are provided as a Source Data file.

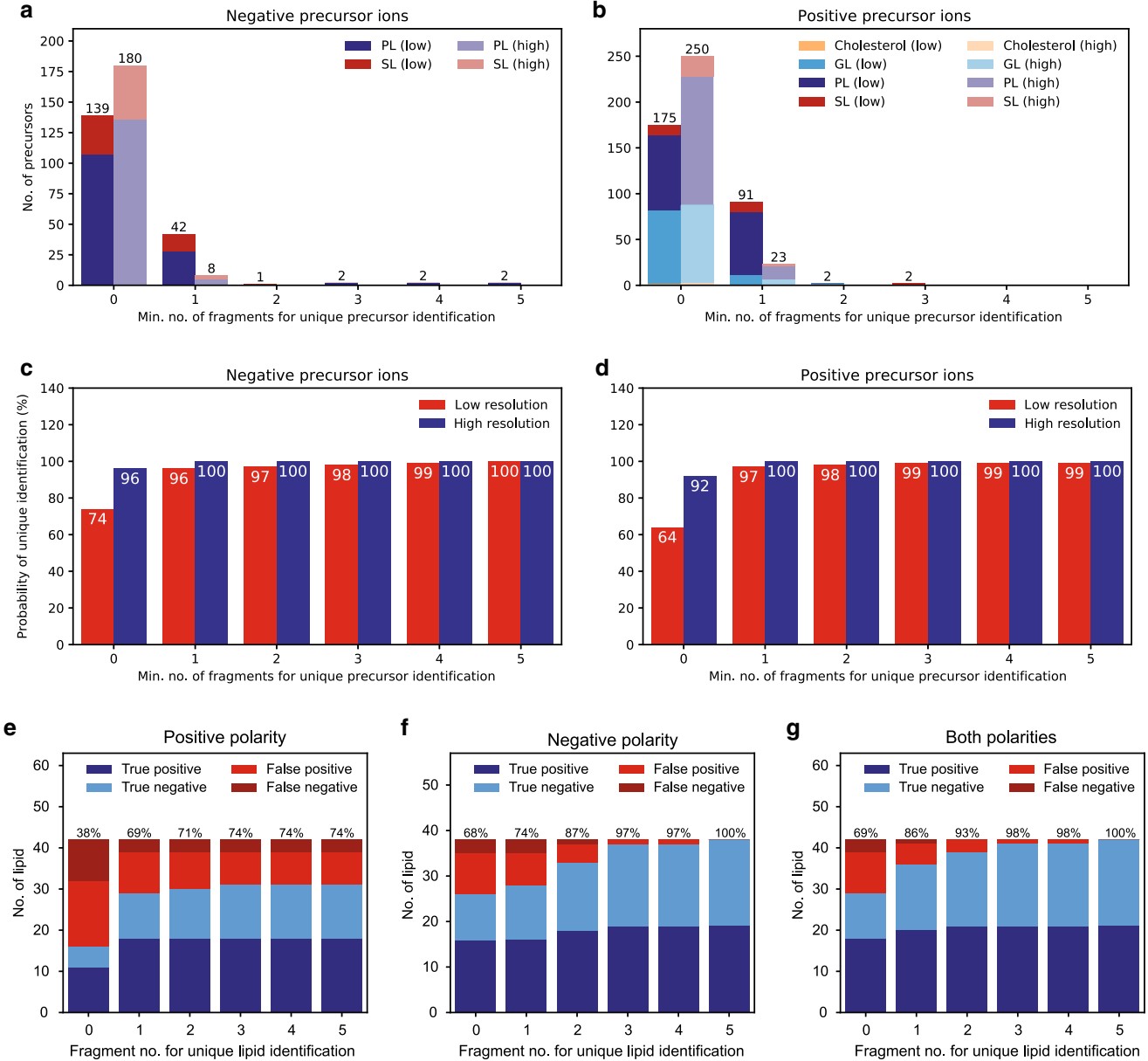

**Fig. 5 Probability and false match analysis proves LipidCreator output as correct.** For probability calculations, the yeast lipidome[34], a set of target lipids and a set of decoy lipids (in total: 188 negative/273 positive lipid ions) were chosen. **a**, **b** The minimum number of lipid ions that can be unambiguously identified using 0 fragments (i.e., precursor mass), 1 arbitrary fragment etc. was calculated in negative and positive ion mode, respectively. The MS and MS/MS mass tolerance was set to ±0.5 Da (representing low resolution instrumentation) or ±2.5 ppm (representing high resolution instrumentation). Sphingolipids (SL) and glycerophospholipids (PL) were investigated in both polarities while, glycerolipids (GL) and sterol lipids (ST) were investigated in positive ion mode only. **c**, **d** Cumulative probability to unambiguously identify any lipid ion when using 0 fragments, 1 arbitrary fragment etc. using 0.5 Da (low) or 2.5 ppm (high) tolerance in positive and negative ion mode, respectively. To verify the calculations, LC/ESI MS/MS experiments were conducted in a yeast lipidome background. The monitored lipid set contained yeast lipids, 21 target lipids, and 21 decoy lipids. Here, (un)identified target lipids are referred to as true positives or false negatives, whereas (un)identified decoys are denoted as false positives or true negatives, respectively. The identification is based on the upper number of used fragments, where 0 fragments means the identification is solely based on precursor mass. The accuracy ratio (%) is plotted above each bar. **e** Numbers of identified and unidentified lipids when considering only positive precursor ions (note that the majority of lipids do not contain more than one fragment in positive mode). Our knowledge base contains only one or two positive fragments for some lipids, therefore the percentage remains unchanged when choosing more fragments. **f** Numbers of identified and unidentified lipids in negative mode. **g** Total number of identified and unidentified lipids when taking both polarities into account. When considering three fragments, all target lipids could be identified and only one decoy was positively identified. Measurements were carried out on a pooled sample of five individual extraction experiments and were conducted in technical triplicates ($n = 3$ independent experimental replicates). Source data are provided as a Source Data file.

in the previous step. Based on our previous results, all lipid species were unambiguous and not common between any of the three sets. For the measurement, we spiked the target lipids into the yeast matrix. We created an LC-ESI PRM assay and

analyzed both target and decoy lipids (See Supplementary Methods for LC-MS/MS parameters). Figure 5e–g illustrates the results. We counted the lipids in each measurement and plotted their numbers. We used the common definition to calculate

accuracy: $acc = (TP + TN) / (TP + FN + FP + TN)$. Correctly identified target lipids (true positives, TP) and correctly unidentified decoys (true negatives, TN) are displayed with blue bars, whereas incorrectly unidentified targets (false negatives, FN) or identified decoys (false positives, FP) are presented with red bars. The results show (i) that LipidCreator provides correct transition lists for lipid identification and (ii) that with increasing fragment number the overall accuracy increases. Especially when we consider the identifications that use up to three fragments (best tradeoff between accuracy and number of used fragments) and both polarities, we identified all target lipids, while we misidentified only one decoy lipid, resulting in an accuracy of 98%. These results are also in line with our theoretical calculations.

**Quantification of plasma lipids and reference material**. To further verify the LipidCreator workflow, we conducted a quantitative lipidomics experiment monitoring 433 lipids in the plasma of 21 healthy Asian human subjects (an overview is given in Fig. 6a, b). For individual lipids, we compared the obtained concentrations to the same set of lipids of a reference material from a mixed American population (NIST SRM 1950) and a recently published ring trial that involved 31 laboratories[51] (Fig. 6a, Supplementary Fig. 9). The rationale behind this was to see if the generation of a list including internal standards and endogenous lipids by LipidCreator leads to a similar quantitative set of lipids as reported in the NIST SRM 1950. All targeted lipids were detected in our control group and in the NIST reference material. The quantitative data reveal that, in the majority of cases, the measured lipid levels are close to the reference material itself. However, a significant proportion of lipid species display a concentration level different from the one reported in the ring trial (Supplementary Fig. 9). This difference might reflect the complex interaction of plasma lipidome, ethnicity, diet and life style[14,52] but would also be caused by sample collection as well as the chosen quantification strategy (Supplementary Fig. 9). The overall lipid distribution of quantifiable lipids is depicted in Fig. 6b.

**Verification of true responses with calibration curves**. In order to prove that LipidCreator computes correct lipid ion signatures that can be the basis of a targeted lipidomics assay, we measured the response of polyunsaturated fatty acyl phospholipids (PUFA-PL) in human platelets matrix. For well selected ion signature, a linear correlation between mass spectrometric response and concentration should be observed. We investigated the fatty acyl combination 18:0–20:4 in PA, PC, PE, PG, PI, and PS lipid classes, which are endogenous lipid species in the majority of mammalian samples and are important pool educts for lipid signaling[53,54]. To create the calibration curve, increasing concentrations were prepared and analyzed by LC/ESI SRM as specified in the Supplementary Methods. We observed a linear response for all analyzed lipids (Supplementary Fig. 2) with an average correlation coefficient of 0.99. This result underlines that the transitions generated with LipidCreator are well suited for targeted lipidomics.

**Validation through lipid signaling in human platelets**. For further validation of LipidCreator, we investigated lipid mediators and their precursor ions during human platelet activation. Platelets are essential for maintaining vascular integrity and hemostasis and are further critically involved in vascular inflammation, as well as acute arterial thrombotic occlusion following the rupture of atherosclerotic plaques[55,56]. Lipid mediators have important functions in platelet physiology, signaling

and energy production[57]. To follow the generation of lipid mediators upon platelet activation and their arachidonate (fatty acyl 20:4, AA)-containing precursors in individual human subjects (Supplementary Table 5), we analyzed platelet signaling in a targeted lipidomics assay. In this assay lipid mediators and AA-containing glycerophospho- (PL), glycerol- (GL), and cholesteryl ester (ChE) were included. Next, we verified the results by spectral matching using a LipidCreator-generated in silico spectral library (Supplementary Figs. 3–8), and determined all lipid concentrations based on the individual responses of endogenous lipids and 19 internal standards. The network analysis based on our results indicates dose-dependent regulation of the lipidome with strong lipid-lipid correlation in distinct clusters of PC, PE, PI, and PA (Fig. 7a). In individual stimulations, lipid species in the PI cluster were dramatically reduced, while the PC and PE clusters were largely unaffected. This result can be interpreted in two ways: first, that the AA-containing PIs were the main contributors to AA generation by the action of lipases, such as phospholipase A2, on lipid precursors; second, that the re-esterification towards PI was slower than that of other PL classes. Therefore, PI metabolism remains the main contributor to the AA pool (Fig. 7b–e) and the subsequent enzymatic oxidation of fatty acyls catalyzed by lipoxygenases (LOX), cyclooxygenases (COX) or cytochrome P450 (CYP)[58,59]. In addition, we observed re-esterification of AA into lipid classes, such as DG and PA, during activation. Most mediators were upregulated in a dose-dependent manner and secreted upon platelet activation. Compared to thrombin activation, CRP had a stronger effect on the regulation of AA lipids and the production and secretion of mediators. Compared to enzymatically generated mediators, such as 15-HETE, $PGD_2$, $TXB_2$, and 12-HETE, the vast amount of mediators downstream of AA were non-enzymatically generated (11-HETE). Applying LipidCreator to platelet biology not only simplified rapid development of the method but also facilitated quantitative experiments to follow the lipid mediator signaling in platelets, further underlining its effectiveness and versatility.

## Discussion

With LipidCreator, we introduced a knowledge-based tool to design targeted lipidomics assays. With its user-friendly graphical interface and supported by built-in interactive tutorials, lipid researchers with any level of experience can quickly learn how to design and apply such assays for their own studies. The user can customize a targeted assay by adding new fragments for lipid species, by introducing heavy labeled isotopes or by enabling CE optimization for different MS platforms. The CE optimization and relative fragment intensity prediction can be extended by users to cover other MS platforms.

We demonstrated the applicability of the assays designed with LipidCreator using multiple computational and experimental methods. With computational simulation, we determined the minimum number of targeted fragments to unambiguously identify a lipid with a certain probability and demonstrated experimentally that for unvalidated assays, multiple fragments and both polarity modes are required for unequivocal identification of lipid species in simple model systems such as yeast. We showed the accuracy of the transition lists generated with LipidCreator by performing an analysis of human plasma lipidome samples and NIST SRM 1950 standard material, which we compared to a recently reported ring trial that used the same reference material[51]. As the ring trial was intended to identify the metrological questions and gaps that currently make it extremely complicated to reach inter-laboratory comparability, the calculated consensus neither reflects the most precise value of a certain analyte nor the true biological quantity. In light of this, we tried to

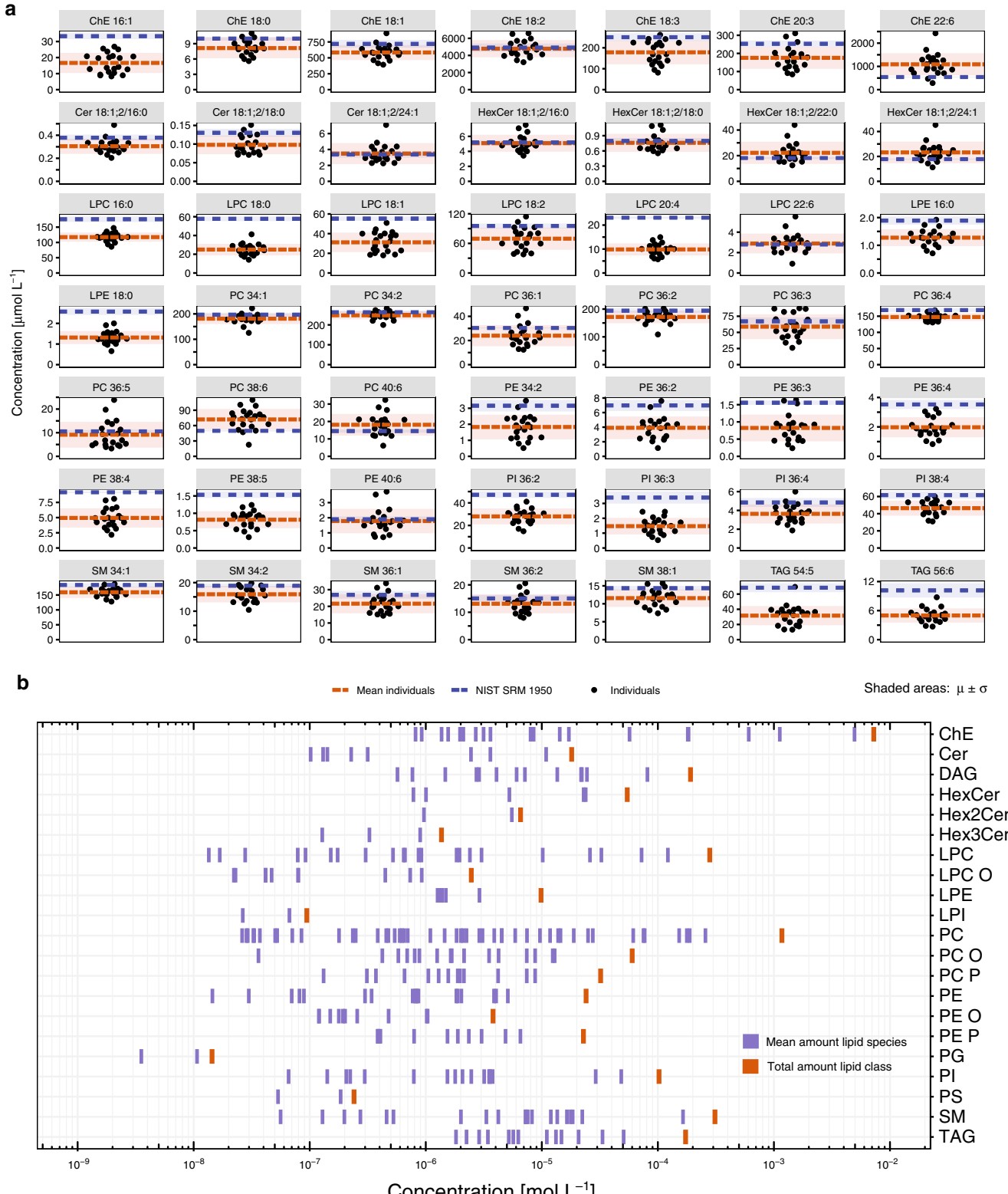

**Fig. 6 Quantitative comparisons in the human plasma lipidome confirms LipidCreator output.** Lipid molecular species from major classes were quantified from the plasma samples of 21 healthy individuals (*n* = 21), and the NIST SRM 1950 reference material, by using transitions generated by LipidCreator. **a** Concentration differences of selected lipid species between healthy individuals (black dots) and average NIST 1950 plasma samples analyzed in this study (blue line). Each sample was measured with four independent technical replicates. 276 lipid species passed QC filters with a linear response $R^2 > 0.8$ and CV < 20%. Shaded areas represent standard deviation around the mean of the individuals and the mean of the NIST 1950 plasma samples. **b** The plasma lipid species concentrations across 22 lipid classes are displayed as blue bars. The sum of the concentrations of individual lipid species of the lipid classes are indicated as vertical thick red bars (right). Source data are provided as a Source Data file.

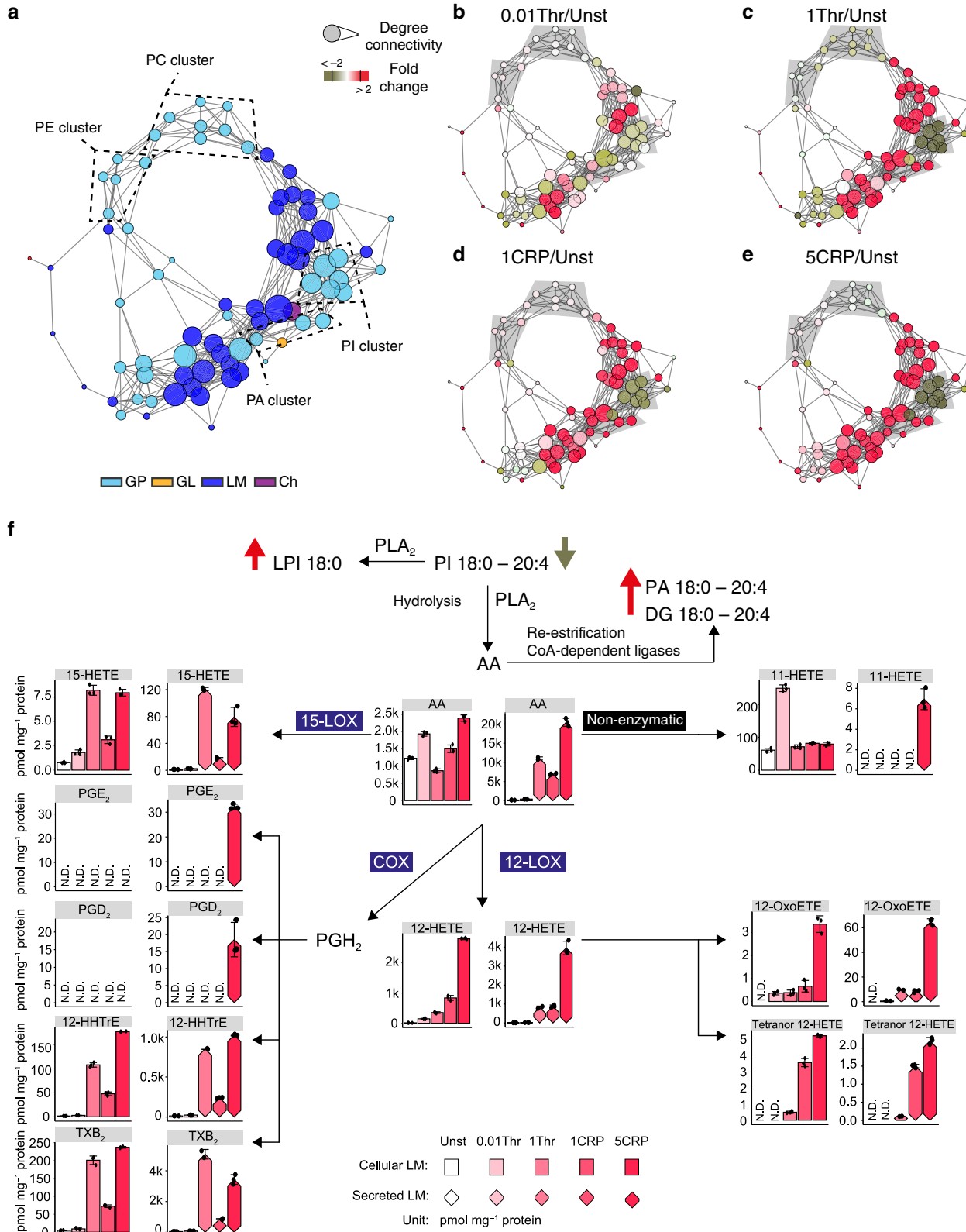

isolate systematic errors due to methodical differences between the intra-laboratory comparison and the reported control group by including the NIST SRM 1950 standard as a reference in our measurements. Here, a direct correlation analysis reveals a good agreement ($R^2 = 0.98$) between the American reference material and our Asian control group. However, the direct comparison of both our control group ($R^2 = 0.79$), as well as of our reference

standard measurements ($R^2 = 0.8$) with the ring trial median of means values reveal discrepancies specific to certain lipid species which we attribute to methodical differences in lipidomics workflows and quantification strategies. The true consensus is likely to be determined using lipid species-specific standards[60] and harmonization efforts[61], which are not yet globally available but are under active development by the International Lipidomics

**Fig. 7 Lipid regulation during human platelet activation. a** Network visualization of the lipid-lipid correlations. Nodes are lipid species. Node size represents the degree of connectivity, and node color represents the analyzed lipid class (see inset). Edges are correlations with r ≥ 0.85. **b-e** Color-coded nodes in the network show the lipid fold change with respect to resting platelets during activation by 0.01 U mL$^{-1}$ thrombin (0.01 Thr), 1 U mL$^{-1}$ thrombin (1 Thr), 1 µg mL$^{-1}$ collagen-related peptide (1 CRP) or 5 µg mL$^{-1}$ CRP (5 CRP); red indicates a fold change greater than or equal to 2, and olive green indicates a fold change less than or equal to 0.5. Data are combined from five independent biological experiments ($n = 5$), and mean values are shown. **f** Arachidonic acid (AA)-based mediator production and release upon stimulation. Bar graphs display the determined mediators in platelet cells. Bar graphs with diamond shape display the secreted mediators. The absolute quantities are reported in pmol mg$^{-1}$ protein. Error bars are presented as the standard deviation of the mean ($n = 3$ independent experimental replicates). 12-lipoxygenase (12-LOX). 15-lipoxygenase (15-LOX). cyclooxogenase (COX). phospholipase A2 (PLA$_2$). Unst: unstimulated. not detectable (N.D.). Source data are provided as a Source Data file.

Society (ILS), Lipidomics Standard Initiative (LSI) and Singapore Lipidomics Incubator (SLING). Nonetheless, we were able to use LipidCreator to generate accurate target transitions to access the core lipid population of the ring trial.

Finally, we applied LipidCreator to investigate lipid mediator signaling during human platelet activation. Here, we demonstrated that similar results in human platelet signaling can be obtained compared to recent reports in mice[36], while we can attribute differences to species-specific (mouse vs. human) characteristics in lipid metabolism during platelet activation.

Currently, LipidCreator provides CE models exclusively for lipid mediators. We thus plan to expand our current models to lipid species of other lipid classes. This requires the availability of more structurally diverse lipid standards and CE-dependent measurements of their fragmentation behavior, complemented by the development and application of more sophisticated machine learning models[62,63]. Due to the lack of available standards in general, rare lipidome wide studies for bacteria such as *E. coli* and archaea and the explicit lack of standards for acyl-PG, acyl-PE in *E.coli*[64] and di- and tetra-alkyl ether lipids in archaea[65] the current implementation focuses on eukaryotic model systems. Here, the majority of lipid classes are covered with some exceptions (Supplementary Data 1). However, we offer a direct interaction with the community via the LIFS webportal to expand the covered lipid classes to other model systems. LipidCreator's lipid class and fragmentation rule database is openly encouraging participation in the form of sharing additional rules, with due credit for contributors.

LipidCreator does not provide a dedicated scoring algorithm to determine false discovery rates but supports a knowledge-based selection of transition lists. Thus, it intentionally does not address the problems of identifying coeluting and / or isobaric lipids and peak area integration. Instead, it provides notes and visual hints to the user about potentially interfering transitions (similar precursor mass and fragment mass with equal polarity). Nevertheless, additional information such as retention time, the elution order of lipids, as well as isotope labeled standards, if available, always need to be considered in order to avoid mis-annotations. To the best of our knowledge, due to the lack of isotope-coded standards there are currently no large-scale methods available to cope with occurring stereoisomers. The isomers existence and quantity should be reported individually based on their retention time[35] or summed up and reported as one species as it was done here and in previous studies[14,51]. In the end, the user has to decide whether these issues pose a potential challenge for their analysis regarding a specific chromatographic setup, target MS platform and acquisition scheme.

In conclusion, LipidCreator facilitates the development of customized targeted lipidomics assays taking into account the entire analytical process by providing all necessary information for its optimization. LipidCreator is well integrated into Skyline for small molecules, which makes it a vendor-independent software for fast assay development in targeted lipidomics. It can be easily extended and customized due to its platform independence

and open source codebase. Method development with Lipid-Creator is well documented, user-friendly and will not only foster basic research but also has the potential to pave the way for clinical investigations.

## Methods

**Chemicals.** Formic acid, tert-butyl methyl ether (MTBE), ammonium formate, ammonium acetate, acetate acid (HAc), sodium chloride, sodium bicarbonate, potassium chloride, glucose, disodium phosphate, HEPES, and calcium chloride were purchased from Sigma Aldrich (Steinheim, Germany). The ULC/MS-grade solvents, acetonitrile (ACN), methanol (MeOH) were obtained from Biosolve (Valkenswaard, Netherlands) and isopropanol (IPA) was purchased from Merck (Darmstadt, Germany). Ultrapure water (18 MΩ cm at 25 °C) was used to prepare solutions. Sodium dodecyl sulfate (SDS) was obtained from Roth (Karlsruhe, Germany), tris(hydroxymethyl)-aminomethane (Tris) from Applichem (Darmstadt, Germany), and sodium chloride (NaCl) from Merck (Darmstadt, Germany). Platelets were activated using collagen-related peptide (Richard Farndale, University of Cambridge, United Kingdom) or thrombin from human plasma (Roche, Germany).

One hundred and thirty-six lipid standards (60 mediators, 58 thereof selected after model review) (See Supplementary Methods) were used to study the lipid fragmentation, build optimal collision-energy models, create in silico spectral libraries and spike-in as internal standards. They were purchased from Avanti (Alabaster, AL, USA) and Cayman Chemical (Ann Arbor, Michigan, USA). Lipid fragment masses of mediators were validated with the Metlin database (https://metlin.scripps.edu/)[66].

**Ethical regulations.** All volunteers gave informed consent for blood samples. The platelet study was approved by the institutional ethics committee (270/2011BO1) at University Hospital Tübingen (Germany) and complies with the declaration of Helsinki and good clinical practice guidelines. The collection and use of human plasma samples has been approved by the Institutional Review Board of the National University of Singapore (NUS-IRB N-17-082E and NUS-IRB B-15-094, Singapore).

**Plasma collection.** Blood from 21 healthy individuals (12 males, nine females; 22–44 years old) was obtained by venipuncture into K3EDTA BD Vacutainer tubes. Platelet poor plasma was collected after centrifugation for 10 min at 3850 × g (4 °C) and stored at −80 °C until analysis. The NIST Standard Reference Material for Human Plasma (SRM1950) was purchased from the National Institute for Standards and Technology (Gaithersburg, MD, USA).

**Platelet isolation and stimulation.** Blood from five individual healthy volunteers was collected in ACD-buffer and centrifuged at 200 × g for 20 min. The obtained platelet-rich plasma was added to the modified Tyrode-HEPES (N-2-hydroxyethyl-piperazone-N′2-ethanesulfonic acid) buffer (137 mM NaCl, 2 mM KCl, 12 mM NaHCO$_3$, 5 mM glucose, 0.3 mM Na$_2$HPO$_4$, 10 mM HEPES, pH6.5). After centrifugation at 900 × g for 10 min and removal of the supernatant, the resulting platelet pellet was resuspended in Tyrode-HEPES buffer (pH7.4, supplemented with 1 mM CaCl$_2$). Freshly isolated and resuspended human platelets were stimulated with 0.01 U mL$^{-1}$ thrombin, 1 U mL$^{-1}$ thrombin, 1 µg mL$^{-1}$ CRP or 5 µg mL$^{-1}$ CRP for 5 min. After centrifugation for 5 min at 640 × g at 25 °C, the pellet and supernatant were separated and separately shock frozen in liquid nitrogen.

**Lipid extraction.** Lipid (except fatty acids and their derivatives) extraction was carried out with MeOH and MTBE (See Supplementary Methods). The dried lipid extract was resuspended in 100 µL of IPA/MeOH/CHCl$_3$ (4:2:1, v/v/v) for further MS analysis. To extract fatty acid and their derivatives, additional 20 µL of acetic acid (99.99%, 17.5 M) was added into the sample. After that, the same incubation and centrifugation procedures as described in the previous paragraph were applied. The dried lipid extract was resuspended in 50 µL of MeOH for further MS analysis. Protein pellets were collected and the protein amount was quantified (See Supplementary Methods).

**Targeted LC-MS/MS analysis**. For the reverse-phase liquid chromatography (LC), an UltiMate 3000-system from Thermo Fisher Scientific (Darmstadt, Germany) was employed. The chromatographic separation was performed according to Supplementary Methods on an Ascentis Express C18 main column (150 mm × 2.1 mm, 2.7 μm, Supelco) fitted with a guard cartridge (5.0 mm × 2.1 mm, 2.7 μm, Supelco). Samples were injected with a volume of 5 μL and analyzed in triplicates. The LC was coupled to a Q-Exactive HF (QEx HF) mass spectrometer (Thermo Scientific, Bremen, Germany) or a QTRAP 6500 (Applied Biosystems, Darmstadt, Germany) to use PRM or SRM acquisition mode (See Supplementary Methods).

To validate lipid mediator species identified with the QTRAP6500 (Applied Biosystems, Darmstadt, Germany) instrument, the QEx HF (Thermo Fisher Scientific, Bremen, Germany) was used to perform high resolution MS full scan (HR-FS) and data independent acquisition (DIA) analyses. DIA method preparation and data analysis were performed with Skyline (See Supplementary Methods).

**Direct infusion of lipid standards on QEx HF and QTOF**. All 136 lipid standards were separately diluted to ~1 μM in IPA/MeOH/CHCl3 (4:2:1, v/v/v) with 7.5 mM ammonium acetate. Then they were infused via robotic nanoflow ion source Tri-Versa NanoMate (Advion BioSciences, Ithaca NY, USA) into the QEx HF mass spectrometer (See Supplementary Methods). The PRM inclusion list was adapted for each lipid standard with NCE ranging from 10–60. The acquisition time was 2 min for each measurement. Raw data were inspected with Thermo Xcalibur version 2.8-280502/2.8.1.2806.

Lipid standards were separately diluted to ~10 μM in MeOH with 0.1% formic acid and then infused via syringe into Agilent QTOF 6545 (Agilent Technologies, Waldbronn, Germany). The inclusion list was adapted for each lipid standard with a collision energy from 10–100 V (See Supplementary Methods). The acquisition time was 1 min for each measurement. Raw data were inspected with Agilent MassHunter version 8.0.

**Data processing and analysis**. Additional information on data processing and analysis is reported in the Supplementary Methods and together with the datasets published in MetaboLights.

**CE-dependent relative fragment intensity prediction**. Nonlinear Regression Model: flipR uses nonlinear regression with an extended log-normal kernel to fit fragment-specific dissociation profiles depending on collision energy and scan-relative intensity (relative to all other fragment intensities in the same scan). We trained the regression model separately for each lipid species and its fragments on HCD (QExactive HF) and CID (Agilent QTOF) platforms using the model function (2), yielding different parameters for each such combination.

We use the probability density function of the log-normal distribution:

$$f_X(x|\mu, \sigma) = \frac{1}{x} \cdot \frac{1}{\sigma\sqrt{2\pi}} \cdot \exp\left(-\frac{(\ln x - \mu)^2}{2\sigma^2}\right) \quad (1)$$

with parameters $\mu \in (-\infty, +\infty)$ and $\sigma, x > 0$ to model collision-energy dependent dissociation profiles for lipid fragments. In order to position the parameterized density function into the collision energy coordinate space we introduced an additional parameter $\delta \in R$ for the necessary shift and an additional parameter $s$ for the rescaling of the density function's height to fit that of the CE profile's height:

$$g_X(x|\mu, \sigma, s, \delta) = s \cdot f_X(x + \delta|\mu, \sigma) \quad (2)$$

Nonlinear Regression and Grid Search: We use the R-packages minpack.lm (https://rdrr.io/cran/minpack.lm/), for nonlinear regression based on the Levenberg-Marquardt nonlinear least squares algorithm) and nls.multstart (https://rdrr.io/cran/nls.multstart/) to perform a bounded grid search optimization over the combined parameter range space for each fragment, using the kernel function from Eq. (2) to model the dependence of scan-relative intensity on (normalized) collision energy.

We currently do not include any specific handling of outliers. For some of our instances, we do see drop-offs in scan-relative intensity because of technical variability of the MS platforms, but due to the high number of repeated measurements for every collision-energy step, these do not influence the overall fits significantly.

The minpack.lm package reports different statistics to evaluate the goodness-of-fit of a nonlinear regression model. minpack.lm calculates the AIC value for each parameterized model and nls.multstart then selects the one with lowest AIC among the evaluated parameter combinations. The AIC balances model complexity by penalizing the number of parameters used for fitting and the prediction error to avoid overfitting.

Residuals and Sum-of-Squared residuals: In order to be able to assess fragment model fits, we apply the Shapiro-Wilk test for normality on the standardized residuals, as calculated between the measured relative scan intensities and the predicted ones for each collision-energy step (see Supplementary Fig. 54). However, especially for low and high collision energies (front and tail of the profile), the fit may diverge, thus leading to consistent over- or under-fitting of the experimental values (see Supplementary Fig. 53). These over- and under-fits lead to deviations from the normality assumption of the residual distribution. In case of

the Thermo QExactive HF platform, we often observe an almost constant range of scan-relative intensities for CE below ~18 (normalized collision energy) (see Supplementary Fig. 52). Therefore, the instrument specific configuration in LipidCreator allows the definition of a minimum collision energy above which the software reports the model-based predicted values.

The overall relative residual for 99.7% (μ(Residual)±3 σ) of all models is well below 0.1 (10%) for the QExactive HF platform, with many instances reaching values below 0.03125 (3.125%). The only notable exception here are the ALA-d14 and AA-d8 species measurement, where a weak precursor fragment fit distorts otherwise low residuals. For this platform, 5 and 10 ppm transition extraction lead to virtually identical results, with the exception of LTB4-d4 and 5,6-DiHETE, demonstrating the higher resolution and low intensity variance benefits of the Orbitrap platform. For the QTOF platform, the residuals do not distribute as favorably as for the QExactive HF platform. We attribute this to the much higher variance of fragment intensities, as well as to the different collision technology (CID on QTOF, vs. HCD on QExactive HF). We therefore see relative residuals of mostly below 0.0625 (6.25%) when we include 99.7% (μ(Residual)±3 σ) of all species models, while some outliers reach values between 0.2 (20%) or even above 0.3 (30%). We have nonetheless integrated the QTOF platform into LipidCreator to demonstrate a) the stability of the FIP nonlinear regression, even on rather noisy data, and b) to show that the spectral database derived from the nonlinear model still has a benefit in spectral matching within Skyline, when compared to a bare binary fingerprint comparison of present/absent masses.

In order to compare the models based on 5 ppm and 10 ppm transition extraction, we calculate the sum-of-squared residuals for each model and normalize by the number of data points used for the regression calculation. We correct for the degrees of freedom (4 estimated parameters) minus 1, to report an unbiased value of the mean squared error (MSE) of each individual model (see Supplementary Fig. 55). A smaller MSE value indicates a better estimate, while extraordinarily large values indicate a less favorable fit estimate, e.g., due to a lack of training data or large relative intensity variation in the data between repeated measurements.

For the Thermo QExactive HF platform, we were able to consistently choose the 5 ppm window. For the Agilent QTOF platform, 5 ppm was sufficient in most of the cases. For the insufficient cases, the 5 ppm transition window picked up too few data points to calculate a usable model. We thus selected model parameters based on the 10 ppm transition window data for some model instances and fragments.

We provide plots of the model predictions, residuals, standardized residual quantile-quantile plot and normalized sum-of-squares of the residuals (MSE) in Supplementary Data 5 (QExactive HF) and Supplementary Data 6 (QTOF) for lipid mediators.

LipidCreator settings are saved as a MS platform-specific parameter file located in the data\ce-parameters directory of the LipidCreator installation directory. The general machine-specific configuration is located in the MS instrument table (data \ms-instruments.csv). For a tutorial on how to use the collision-energy calculation, please see Supplementary Note 2 Collision-energy optimization function.

**Reporting summary**. Further information on research design is available in the Nature Research Reporting Summary linked to this article.

## Data availability

All raw files and processed data tables are available from public repositories. Skyline projects for the human platelet activation measurements are available from the Panorama repository [https://panoramaweb.org/lipidcreator.url]. Raw MS data, mzML converted data, transition lists, picked and integrated peak areas exported from Skyline and the final, quantified lipid result tables are available from MetaboLights under the accession codes MTBLS1376 [https://www.ebi.ac.uk/metabolights/MTBLS1376] (yeast data), MTBLS1375 [https://www.ebi.ac.uk/metabolights/MTBLS1375] (targeted analysis of human plasma samples), MTBLS1369 [https://www.ebi.ac.uk/metabolights/MTBLS1369] (human platelet data: targeted LC-MS/MS analysis of phospholipids, glycerolipids and sphingolipids), MTBLS1381 [https://www.ebi.ac.uk/metabolights/MTBLS1381] (human platelet data: targeted analysis of mediators), MTBLS1382 [https://www.ebi.ac.uk/metabolights/MTBLS1382] (human platelet data: DIA validation), MTBLS1333 [https://www.ebi.ac.uk/metabolights/MTBLS1333] (training Data for CE optimization model training of lipid mediators: QExactive HF Platform), MTBLS1334[https://www.ebi.ac.uk/metabolights/MTBLS1334] (training Data for CE optimization model training of lipid mediators: QTOF Platform). Averaged CE spectra of lipid mediator standards measured on the Thermo QExactive HF and Agilent QTOF platforms are available from MassBank at https://massbank.eu/MassBank/Result.jsp?type=rcdidx&idxtype=site&srchkey=ISAS_Dortmund. The source data underlying Figs. 4, 5a–d, 5e–g, 6a,b, 7a–e, f and Supplementary Figure 2 are provided as a Source Data file. All other data are available from the corresponding author on reasonable request.

## Code availability

The scripts underlying Figs. 4–7 and Supplementary Figure 2 are provided as Supplementary Data 7. The source code of LipidCreator is available [https://github.com/lifs-tools/lipidcreator]. The source code of flipR, the training harness and the code required to recreate Supplementary Data 5 and Supplementary Data 6 are available from [https://github.com/lifs-tools/flipr]. A binary compiled version of LipidCreator is

available as Supplementary Software 1. Releases of LipidCreator are available at [https://lifs.isas.de/lipidcreator] and at Zenodo [https://doi.org/10.5281/zenodo.3529484].

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

## Acknowledgements

This project was funded by the BMBF grant LIFS (de.NBI /BMBF 031L0108A,B), granted to R.A. and D.S., the Deutsche Forschungsgemeinschaft (DFG, German Research Foundation) – Projektnummer 374031971 – TRR 240, supported by the Leibniz Association awarded to R.A. and supported by the Ministerium für Kultur und Wissenschaft des Landes Nordrhein-Westfalen and the Regierende Bürgermeister von Berlin, Senatskanzlei Wissenschaft und Forschung - inkl. Wissenschaft und Forschung, and the Bundesministerium für Bildung und Forschung granted to R.A. This work was further supported by the VILLUM Foundation (VKR023439; C.S.E.; http:// villumfonden.dk) and the Lundbeckfonden (R54-A5858; CSE; www. lundbeckfoundation.com) granted to C.S.E., the Skyline R01 ("Skyline Targeted Proteomics Environment" R01 GM103551) granted to B.M. and the National University of Singapore via the Life Sciences Institute (LSI) and the Singapore National Research Foundation (NRFI2015-05) granted to M.R.W. We thank Fernando Martínez-Montañés for providing the yeast samples.

## Author contributions

B.P., D.K., D.S., and R.A. designed the concept of LipidCreator. B.P., C.S.E., M.H., and R.A. discussed and finalized the nomenclature. B.M. and B.S.P. performed the Skyline integration for LipidCreator. R.A., B.P., F.T., B.B., P.I.B., S.H.T., M.Y.C., C.C., S.M., and M.C.M. designed and performed the experiments. B.P., N.H., B.B., C.C., and R.A. analyzed the experiments. N.H. implemented and trained the relative intensity prediction models. B.P. and N.H. validated the model predictions. D.K., N.H., and B.S.P. wrote the source code. N.H. prepared and submitted the datasets to MetaboLights and MassBank. B.P., D.K., C.S.E., D.S., S.M., O.J.S., B.M., O.B., N.H., M.R.W., and R.A. discussed the content. B.P., D.K., N.H., and R.A. wrote the manuscript.

## Competing interests

The authors declare no competing interests.
