## [Peer Review File · Nature Communications]

REVIEWERS' COMMENTS:

Reviewer #2 (Remarks to the Author):

General comments

The authors addressed the majority of my comments on the initial draft of the manuscript. However, the discussion still does not mention lipid species not covered by LipidCreator and a few fixes and improvements to the software itself are required.

Specifically the following comments remain:

1. Manuscript

1. Page 7, line 4: Please explain why "a modified global sequence alignment" was used, instead of simply calculating the percentage of matching fragments.

2. Page 9, lines 29ff: Include a discussion of the considerable reduced coverage of the *E. coli* lipidome and of the specific species not covered in other organisms.

3. Page 11, line 28: Add a reference to the MetLin paper.

4. Page 12, line 4: "To extract fatty acids and their derivatives ..."

5. Page 15, line 24: "... peak areas exported from Skyline ..."

6. References:

- Use consistently either abbreviated or full journal names
- Check references for correctness and completeness (e.g. 16, 30, 37, 39, 43, 45, 50, ...)

7. Figure 1: It is still not clear from looking at the figure, which steps are performed in LipidCreator and Skyline respectively.

Enclose points 1-6 performed in LipidCreator by a single frame in Fig 1A and make a single frame around the points 1-6 performed in LipidCreator and 7-9 in Skyline similar to Fig 1A and add the LipidCreator and Skyline logos respectively.

8. Figure 2: Remove dimmed background images from figure, as they do convey and information.

9. Figure 4: Add total number of lipid species for each of the organisms and add a legend for the bar colors.

10. Figure 7: I would suggest to move this figure into the supplement.

2. LipidCreator Software

1. Although the provided tutorials are helpful, it would be preferable to implement the workflow as a so-called "Wizard", so that the users are guided through it.

2. Element selection in lists (e.g. "Head group") does not adhere to common interface guidelines: Using "Ctrl", single entries should be selected/deselected; using "Shift", all entries between the previously selected and the current element should be selected. "Ctrl-a" should select all elements in the list.

3. Supplementary Material S1

1. S1.1.4: Either remove "extraction" from subheadings or change to "From xxxxx".

2. Figure S2: Justify the rather large 20% CI.

3. S1.3.2: Add the table containing the species from the different data sets, which are not by covered by LipidCreator (page 20 of the rebuttal) and include a comment column with an explanation why they are not covered.

4. Supplementary Material S2

1. Page S2.3: Although the tutorials are indeed helpful, I still see a need to include here a page with the general workflow overview. What steps are necessary in what order; what files have to be created, exported, etc...

5. Supplementary Material S3

1. The quality of the all figures is rather low (especially in S3A). All figures should be included as vector graphics to allow lossless enlargement.

Referee 2

Prof. Dr. Robert Ahrends
PI Lipidomics
Institute of Analytical Chemistry
Währinger str. 38
1090 Vienna
Austria
P: +43 1-4277-52304
M: +49 (0)176.72655208
E: robert.ahrends@univie.ac.at
H: www.lipidomics.at

Vienna, 27th of March 2020

Subject: LipidCreator, NCOMMS-20-04729A former NCOMMS-20-04729-T

Dear referee,

Many thanks for the detailed suggestions on the manuscript, which were very helpful during the revision process. We are very happy to present to you our revised manuscript: "LipidCreator workbench to probe the lipidomic landscape". We addressed the remaining comments and suggestions. We included a Wizard to guide the unexperienced user through LipidCreator, applied further usability fixes, such as enabling multiple selection of lipid head groups in our software. We discussed the lipid coverage of non-eukaryotic species briefly in the discussion section of the manuscript and added the missing species to Supplementary Table 1. The limited coverage of *E.coli* lipid species is due to the fact that for those lipid classes no commercial standards are available, that these classes were not validated by our own measurements and that the MS based literature on them is quite sparse. Therefore, we decided not to include them in this first release of LipidCreator. However, to further reduce the species, which we do not cover we included additional lipid classes in the sterol lipid category. We address the remaining requests and suggestions point by point below in the response to the reviewer. We are convinced that LipidCreator is now covering all necessary features, and therefore the offered user support in conjunction with the tool will be of great interest to a broad readership addressing the urgently needed requirements for rapid targeted lipidomics assay development. We hope that you agree on this and are considering the manuscript now ready for publication in Nature Communications. Thank you again for all your help and support.

Sincerely,

Robert Ahrends

Reviewer #2 (Remarks to the Author):

General comments

The authors addressed the majority of my comments on the initial draft of the manuscript. However, the discussion still does not mention lipid species not covered by LipidCreator and a few fixes and improvements to the software itself are required.

Specifically the following comments remain:

1. Manuscript

1. Page 7, line 4: Please explain why "a modified global sequence alignment" was used, instead of simply calculating the percentage of matching fragments.

Our method uses a probability function with a mass accuracy of 5ppm (+/-2.5). It compares the fragment masses within this accuracy range via modified global sequence alignment to infer the optimal fragment assignment between any two of the lipid fragments that are compared to calculate the minimum number of fragments for unambiguous identification. This method avoids that any two fragments from one list are assigned to the same fragment from the other list.

2. Page 9, lines 29ff: Include a discussion of the considerable reduced coverage of the *E. coli* lipidome and of the specific species not covered in other organisms.

Based on the publications of Matyash et al. (2008) and Herzog et al. (2012) we achieve a coverage of 100% on the wild type *E. coli* lipidome. However, if the *E. coli* metabolism is perturbed, new lipid classes can occur, as reported by Jeucken et al. (2019). For these lipid classes, so far no commercial standards are available and these lipid classes have not been validated by other labs. In order to increase the coverage, we have added the following sterol lipids and their corresponding fatty acyl esters: Ergosterol, Ergostadienol, Desmosterol, Lanosterol, Stigmasterol. We added the following paragraph:

„Due to the lack of available standards in general, rare lipidome wide studies for bacteria such as *E. coli* and archaea and the explicit lack of standards for acyl-PG, acyl-PE in *E.coli*⁶⁴, di- and tetra-alkyl ether lipids in archaea⁶⁵ the current implementation focuses on eukaryotic model systems. Here, the majority of lipid classes are covered with some exceptions (Supplementary Data 1).“

Please note that we moved Supplementary Table 1 to Supplementary Data 1.

3. Page 11, line 28: Add a reference to the MetLin paper.

We added this reference.

4. Page 12, line 4: "To extract fatty acids and their derivatives ..."

We corrected this mistake.

5. Page 15, line 24: "... peak areas exported from Skyline ..."

We corrected this mistake.

6. References:

- Use consistently either abbreviated or full journal names
- Check references for correctness and completeness (e.g. 16, 30, 37, 39, 43, 45, 50, ...)

We thank the reviewer for spotting our oversight. We have updated the references accordingly.

7. Figure 1: It is still not clear from looking at the figure, which steps are performed in LipidCreator and Skyline respectively. Enclose points 1-6 performed in LipidCreator by a single frame in Fig 1A and make a single frame around the points 1-6 performed in LipidCreator and 7-9 in Skyline similar to Fig 1A and add the LipidCreator and Skyline logos respectively.

We added two frames to better distinguish LipidCreator and Skyline based features.

8. Figure 2: Remove dimmed background images from figure, as they do convey and information.

We have removed the background images.

9. Figure 4: Add total number of lipid species for each of the organisms and add a legend for the bar colors.

We modified Figure 4 and added the total number of lipid species to each subplot. The figure now also has a color legend to identify the lipid categories.

10. Figure 7: I would suggest to move this figure into the supplement.

The results of the figure are discussed in the main text and are an essential part of the validation. Therefore we politely disagree here with referee two and would like to keep the figure as a part of the main manuscript.

2. LipidCreator Software

1. Although the provided tutorials are helpful, it would be preferable to implement the workflow as a so-called "Wizard", so that the users are guided through it.

We agree with the referee, a wizard guiding the user through the workflow can be useful, especially for novice users or users with a very focused application goal. Therefore, we have implemented a wizard interface as per the referee's request. The user can start the wizard via the menu: File -> Run Wizard and experience the magic of LipidCreator. The wizard is document in the Supplementary Information. A small selection of panels is provided here:

2. Element selection in lists (e.g. "Head group") does not adhere to common interface guidelines: Using "Ctrl", single entries should be selected/deselected; using "Shift", all entries between the previously selected and the current element should be selected. "Ctrl-a" should select all elements in the list.

Thank you for the clarification. Indeed, it is a useful and a convenient feature. We have now enabled it for all list boxes (containing e.g. PL head groups, mediators) in LipidCreator. Ctrl-a now selects all elements in the list boxes.

3. Supplementary Material S1

1. S1.1.4: Either remove "extraction" from subheadings or change to "From xxxxx".

We have removed "extraction" from the subheadings.

2. Figure S2: Justify the rather large 20% CI.

We are following the recommendations of the U.S. Food and Drug Administration, Guidance document: Bioanalytical Method Validation Guidance for Industry, Issued by: Center for Drug Evaluation and Research, 2018, page 20.

URL: <https://www.fda.gov/regulatory-information/search-fda-guidance-documents/bioanalytical-method-validation-guidance-industry>.

3. S1.3.2: Add the table containing the species from the different data sets, which are not by covered by LipidCreator (page 20 of the rebuttal) and include a comment column with an explanation why they are not covered.

We added further lipid classes to LipidCrator (sterols) and therefore we are supporting more lipid classes then in the earlier reviewed version. The unsupported lipid species are now also added with a justification to the Supplementary Data 1.

Unsupported lipid classes:

Mouse heart: GM1, AcylCarnitine, Ubiquinone;

E.coli: 3 x DLCL, 15 x aPG, 6 x aPE

4. Supplementary Material S2

1. Page S2.3: Although the tutorials are indeed helpful, I still see a need to include here a page with the general workflow overview. What steps are necessary in what order; what files have to be created, exported, etc...

We have added the following workflow overview figure to the Supplementary Information document as Supplementary Figure 10. We describe the workflow in the Supplementary Note 2. We have also added a walkthrough that demonstrates the new wizard. This offers a simplified usage of LipidCreator, while at the same time allowing advanced users to access and modify all features manually.

5. Supplementary Material S3

1. The quality of the all figures is rather low (especially in S3A). All figures should be included as vector graphics to allow lossless enlargement.

Due to the upload size limitation, we were previously not able to upload the full resolution documents. We now provide the Supplementary Data 5 (formerly 3A) and 6 (formerly 3B) in better resolution.